Compi: a framework for portable and reproducible pipelines

López-Fernández Hugo 1 2 3
Graña-Castro Osvaldo 4
Nogueira-Rodríguez Alba 1 2 3
Reboiro-Jato Miguel 1 2 3
Glez-Peña Daniel dgpena@uvigo.es 1 2 3
1 ESEI: Escuela Superior de Ingeniería Informática, University of Vigo , Ourense , Galicia , Spain
2 CINBIO - Centro de Investigaciones Biomédicas, University of Vigo , Vigo , Galicia , Spain
3 SING Research Group, Galicia Sur Health Research Institute (IIS Galicia Sur). SERGAS-UVIGO , Vigo , Galicia , Spain
4 Bioinformatics Unit, Structural Biology Programme, Spanish National Cancer Research Centre (CNIO) , Madrid , Madrid , Spain
Winkler Robert
Electronic publication date: 2021 Jun 18
Publication date: 2021
Volume: 7
Electronic Location ID: e593
Received 2021 Jan 28; Accepted 2021 May 21
Copyright: ©2021 López-Fernández et al.
Copyright year: 2021
Copyright holder: López-Fernández et al.
License: This is an open access article distributed under the terms of the Creative Commons Attribution License, which permits unrestricted use, distribution, reproduction and adaptation in any medium and for any purpose provided that it is properly attributed. For attribution, the original author(s), title, publication source (PeerJ Computer Science) and either DOI or URL of the article must be cited.
License URL: https://creativecommons.org/licenses/by/4.0/

Keywords: Computational pipelines, Workflow management systems, Application development framework

Funding: Consellería de Educación, Universidades e Formación Profesional (Xunta de Galicia) ED431C2018/55-GRC Ministerio de Economía, Industria y Competitividad, Gobierno de España under the scope of the PolyDeep project DPI2017-87494-R Xunta de Galicia ED481A-2019/299 This work was supported by the Consellería de Educación, Universidades e Formación Profesional (Xunta de Galicia) under the scope of the strategic funding ED431C2018/55-GRC Competitive Reference Group and Ministerio de Economía, Industria y Competitividad, Gobierno de España under the scope of the PolyDeep project (DPI2017-87494-R). Alba Nogueira-Rodríguez is supported by a pre-doctoral fellowship from Xunta de Galicia (ED481A-2019/299). There was no additional external funding received for this study. The funders had no role in study design, data collection and analysis, decision to publish, or preparation of the manuscript.

==============================
Compi is an application framework to develop end-user, pipeline-based applications with a primary emphasis on: (i) user interface generation, by automatically generating a command-line interface based on the pipeline specific parameter definitions; (ii) application packaging, with compi-dk, which is a version-control-friendly tool to package the pipeline application and its dependencies into a Docker image; and (iii) application distribution provided through a public repository of Compi pipelines, named Compi Hub, which allows users to discover, browse and reuse them easily. By addressing these three aspects, Compi goes beyond traditional workflow engines, having been specially designed for researchers who want to take advantage of common workflow engine features (such as automatic job scheduling or logging, among others) while keeping the simplicity and readability of shell scripts without the need to learn a new programming language. Here we discuss the design of various pipelines developed with Compi to describe its main functionalities, as well as to highlight the similarities and differences with similar tools that are available. An open-source distribution under the Apache 2.0 License is available from GitHub (available at https://github.com/sing-group/compi). Documentation and installers are available from https://www.sing-group.org/compi. A specific repository for Compi pipelines is available from Compi Hub (available at https://www.sing-group.org/compihub.

Introduction

Bioinformatics units routinely deal with massive data analyses, which require combining multiple sequential or parallel steps using specific software tools (Perkel, 2019). Many of these computational pipelines are published regularly in the form of protocols, best practices or even fully runnable pipelines. They implement all the required steps and dependencies in order to ensure the reproducibility of the analyses and facilitate job automation (Grüning et al., 2018). Thus, scientific computational pipelines must provide three key features: reproducibility, portability and scalability. Container technologies such as Docker or Singularity are the most widely used tools to ensure that pipelines run in stable environments (i.e., always using the exact same version of pipeline dependencies) and make it easy to run the pipeline on multiple hardware platforms (e.g., workstations or cloud infrastructures), enforcing reproducibility and portability. Moreover, scalable pipelines must support running on HPC (High Performance Computing) resources using cluster management and job scheduling systems such as Slurm or SGE.

For the above reasons, a wide variety of workflow management systems have been released in recent years that address these issues in different ways. Tools with graphical user interfaces, such as Galaxy (Afgan et al., 2018), are designed for scientists with little or no programming experience, although such tools can be difficult to set up and configure. Furthermore, command-line based applications such as Nextflow (Di Tommaso et al., 2017), Snakemake (Köster & Rahmann, 2012), or SciPipe (Lampa et al., 2019), provide feature-rich workflow engines oriented to bioinformaticians with medium-to-high programming skills. The Common Workflow Language (CWL; https://www.commonwl.org/) definition represents another alternative, since it defines a specification and offers a reference implementation, but does not provide a complete framework (Leipzig, 2017). Other frameworks like Galaxy or Taverna made significant progress to support the execution of workflows defined in CWL and other tools allow to export their workflows into CWL (e.g., Snakemake) or import them from CWL.

Despite the existence of such remarkable workflow management systems, scientists with basic scripting skills (e.g., able to create shell scripts invoking command-line tools) but lacking advanced programming skills (e.g., knowledge of programming languages such as Python or Go), are usually overwhelmed due to the high complexity of these systems, and could be hampered to use or create their own workflows. In this sense, tools such as Bpipe (Sadedin, Pope & Oshlack, 2012) help to assemble shell scripts into workflows to aid in job automation, logging and reproducibility.

Compi is specially designed for researchers who want to take advantage of common workflow engine features (e.g., automatic job scheduling, restart from point of failures, etc.) while maintaining the simplicity and readability of shell scripts, without the need to learn a new programming language. Nevertheless, Compi also incorporates several features that meet the needs of the most advanced users, such as support for multiple programming languages or advanced management and control of workflow execution. In this sense, Compi is more than a workflow engine, it was created as an application framework for developing pipeline-based end-user applications by providing: (i) automatic user interface generation—generates a classical command-line interface (CLI) for the entire pipeline based on its parameter specifications; (ii) application packaging—provides a version-control-friendly mechanism to package the pipeline application and its dependencies into a Docker image; and (iii) application distribution, supported by Compi Hub (Nogueira-Rodríguez et al., 2021)—a public repository where researchers can easily and freely publish their Compi pipelines and related documentation, making them available for other researchers.

Compi has been adopted by our research group to create pipelines for multiple research projects where other systems were not appropriate enough (e.g., the creation of complex pipelines for phylogenomics or training of models based on deep learning for image classification). Likewise, pipelines developed with Compi in collaboration with other research groups have already been published. These are: Metatax, a pipeline to analyze biological samples based on 16S rRNA gene sequencing (Graña Castro et al., 2020a; Graña Castro et al., 2020b), FastScreen, a pipeline for inferring positive selection in large viral datasets (López-Fernández et al., 2020), and GenomeFastScreen, an extension of the FastScreen pipeline (López-Fernández et al., 2021).

Materials & Methods

This section depicts the most relevant technical details regarding the implementation of Compi. An open-source distribution under the Apache 2.0 License is available from GitHub (https://github.com/sing-group/compi). Documentation and installers are available from https://www.sing-group.org/compi.

XML for pipeline definition

Compi pipelines are defined in a single XML (eXtensible Markup Language) file that includes: (i) tasks and dependencies between them, (ii) pipeline input parameters that are forwarded to tasks, and (iii) metadata, including parameters and tasks descriptions, which are useful for automatic generation of the user interface, as well as for pipeline documentation.

We have chosen XML instead of JSON, YAML, or a custom DSL (Domain Specific Language) to reconcile the various requirements simultaneously. First, we wanted a high degree of interoperability. XML, JSON, and YAML can be easily generated and parsed in almost any programming language, although YAML can pose portability issues in some cases as YAML parsers in different languages can produce different results. In contrast, a DSL (Di Tommaso et al., 2017) is less interoperable, being difficult to produce or consume from languages other than the one on which the DSL is based. Second, XML is appropriate for dealing with long chunks of text, such as the embedded source code for pipeline tasks. Since Compi is language agnostic, these tasks can be defined in any programming language. Thanks to the XML CDATA (character data) blocks, which allow declaring a section of the XML that should not be parsed as XML, it is possible to include source code without any alteration. Embedding task code in JSON is virtually not feasible, as tabs and line breaks, which are key characters in languages like Python, must be escaped in JSON files. In YAML, however, it is easier thanks to its multiline literals. In DSLs it depends on whether the programming language the DSL is based on allows you to declare multiline strings easily, as Python does. Third, XML is easy to validate syntactically and semantically through schemas, which are also present in JSON and YAML. DSLs take advantage of the language parser the DSL is based on. Fourth, for security reasons, since Compi is designed to run pipelines defined by third-party programmers, YAML is a less secure language in this regard, as runnable code could be embedded in fields that were not intended for this purpose (see https://www.arp242.net/yaml-config.html). Fifth comes readability. YAML is the clearer winner in this regard, as readability is the key feature of this format. Although XML is less readable than YAML, the comparison with JSON and DSLs is a more subjective matter. Based on these five requirements, the choice was between XML and YAML: while XML is more secure and portable, YAML is more readable. Finally, we selected XML, prioritizing its security, portability and popularity. Also, while writing XML files could be verbose, YAML is syntactically aware of whitespace, which is a welcome feature in the Python community, but still debated outside of it.

The workflow execution engine

The workflow execution engine of Compi is implemented purely in Java and is responsible for multi-thread task scheduling, monitoring, and standard error and output logging of tasks. First, it computes the DAG (Direct Acyclic Graph) of the task dependencies, since a task may depend on a set of tasks that must be run before the given task. Right after starting, or whenever a task finishes its execution, the engine reacts and all the tasks that can be run, i.e., the tasks whose dependencies are now complete, are sent to a worker thread pool that has a parameterizable size. When there are no more tasks to run, the pipeline execution ends.

Every time a task is about to run and there is a free thread in the pool, a new subprocess is spawned by invoking the system Bash interpreter to execute the task script. Pipeline parameters are passed to the task script through environment variables, which is a robust, standard mechanism with two main benefits. On the one hand, environment variables are easily accessed via “$variable_name” or more complex expressions, so pipeline parameters are available directly within the task script. On the other hand, this allows Compi to pass parameters to scripts written in languages other than Bash, because virtually all programming languages give access to environment variables.

The way to execute languages differently from Bash is through task interpreters. Any task can have a task interpreter defined in the pipeline specification, which is an intermediate, user-defined bash script intended to take the task script as input and call an interpreter from a different programming language.

Moreover, it is possible to define task runners, which are the same concept as interpreters, but with a different purpose. They are not defined within the pipeline specification, but rather at execution time, and are intended to tailor task execution to specific computing resources, without modifying the pipeline itself. In this way, the workflow definition is decoupled from the workflow execution, making it possible to change the way tasks are executed without modifying the workflow XML file. For instance, if the tasks must be run in a cluster environment such as SGE or Slurm, computations must be initiated via a submission command (qsub in SGE or srun in Slurm). Task runners intercept task execution by changing the default Bash interpreter with queue submissions.

Compi project architecture

Compi comprises three main modules. The most important is the core module, which contains the workflow execution engine, with its main data structures. On top of it, there are two additional modules. On the one hand, the cli (command-line interface) module contains the command-line user interface for running pipelines, which generates a specific pipeline application tailored for pipeline tasks and parameters. On the other hand, the dk (Development Kit) module allows to create a portable application in the form of a Docker image and publishing the pipelines at Compi Hub.

Compi Hub

As discussed above, Compi Hub is a public repository where Compi pipelines can be published. The Compi Hub front-end was implemented using the Angular v7 web application framework, while the back-end was implemented using TypeScript and offers a RESTful API that supports all the functionality of the front-end. This REST API is also used by the compi-dk tool to allow pipeline developers to publish their pipelines from the command line. The Compi Hub back-end runs in a Node.js server and uses a MongoDB database to store the data.

Results

Compi can be seen as an ecosystem comprising: (i) compi, the workflow engine with a command-line user interface to control the pipeline execution, (ii) compi-dk, a command-line tool to assist in the development and packaging of Compi-based applications, and (iii) Compi Hub, a public repository of Compi pipelines that allows users to easily discover, navigate and reuse them (Nogueira-Rodríguez et al., 2021). Figure 1 illustrates the development and deployment lifecycle of Compi-based applications. A Compi-based application is a CLI application in which a user can run a pipeline in whole or in part by providing only the parameters of each task. This CLI displays all parameter names and descriptions following standard CLI conventions to help users use the pipeline application more easily.

Figure 1 Lifecycle of developing, packaging, distributing, and running Compi-based applications.

For developers, this lifecycle comprises three main stages: pipeline development (including testing the pipeline as pipeline user), application packaging, and distribution.

Compi helps pipeline developers by automatically determining the parameters required by a pipeline. This implies, on the one hand, that comprehensive help is generated for the user on how to use the application and, on the other hand, that the application parameters are automatically parsed when the pipeline is executed. This is complemented by several built-in features added to the CLI application, such as an advanced task execution control, allowing to run only specific tasks in a pipeline, or log management support per task. Therefore, building a CLI application from a Compi pipeline is a straightforward process that allows Compi users to focus on pipeline development. The “Pipeline design” section explains how Compi users can design their pipelines.

Once the pipeline is developed, it can take two complementary paths: it can be executed directly with compi (top right of Fig. 1) or it can be packaged as a portable end-user CLI application in a Docker image using compi-dk (application packaging area in Fig. 1), which can be then executed via Docker. Pipeline execution and dependency management are explained in the “Pipeline execution and dependency management” section, while the application packaging in Docker images is explained in the “Reproducible Application Packaging with Docker” section.

When an end-user pipeline application is ready to be published, Compi provides pipeline developers with a distribution platform called Compi Hub, where they can share different versions of their pipelines along with usage documentation, example datasets, parameter values, as well as links of interest (e.g., Docker Hub, Github). Community users can then browse the Compi Hub for pipelines and can explore the helpful documentation generated by the authors of each pipeline, as well as other information automatically generated by the platform, such as an interactive DAG representation of the pipeline. This topic is covered in the “Pipeline distribution via Compi Hub” section.

Finally, it is important to note that compi-dk projects are particularly designed to be compatible with version control systems, as the required configuration and pipeline files are text files. Dependency management via a Dockerfile is key to achieve this, complemented by Compi-specific dependency management performed by compi-dk (e.g., the compi executable added to the Docker image of the pipeline). Therefore, a compi-dk project only requires that compi, compi-dk and Docker are installed in order to be built or run. With this feature, Compi gives pipeline developers the ability to use version control systems to keep their pipeline safe and to version their code.

Pipeline design

As explained previously, Compi pipelines are defined in an XML document that includes user parameters, tasks, and metadata. Figure 2 shows an example of a minimal pipeline. This sample pipeline defines two parameters (“name” and “output”), described in the corresponding “params” section, and two tasks (“greetings” and “bye”) described in the “tasks” section. The parameter descriptions are used to automatically generate both the user interface and the documentation for the pipeline. In the same way and for the same purpose, tasks are described in the “metadata” section, which allows the pipeline developer to describe them in a human-readable manner.

Figure 2 Minimal pipeline example of a Compi pipeline with two tasks.

A specific section (“params”) defines two parameters (“name” and “output”), used in the two pipeline tasks, named “greetings” and “bye”. Both tasks print the value of the “name” parameter to the path specified in the “output” parameter. By default, tasks are executed as Bash commands and this is the case of the “greetings” task. The “bye” task is written in Perl and the “interpreter” parameter of the task definition indicates how to invoke the Perl interpreter to run the task code.

Tasks are defined as “task” elements within the “tasks” section. The main components of a task are: source code, parameters and dependencies. The source code is placed inside a “task” element, and the parameters used by a task are defined within the “params” attribute, and those tasks that the current task depends on are defined in the “after” attribute. For instance, in Fig. 2, the “bye” task, uses the “name” and “output” parameters, and depends on the “greetings” task. Parameter values are passed to the tasks as environment variables, which is a standard method that any programming language gives access to. In this way, the Compi workflow execution engine does not need to process the task code in any way (e.g., to perform variable substitution), guaranteeing the possibility of using any programming language to define the code of the task. In this sense, the task code is written in Bash by default, although other scripting languages (e.g., Python, R, AWK, etc.) can be used under a suitable interpreter through an “interpreter” parameter. For example, in Fig. 2, the “bye” task is written in Perl and the “interpreter” parameter indicates how to invoke the Perl interpreter to run the task code.

A special type of tasks are parallel iterative tasks (or loop tasks), defined via “foreach” elements, which spawn multiple parallel processes with the same code on a collection of items. This collection of items is provided by a user-specified source of items, which could be a comma-separated list of values, a range of numbers, the files in a specific directory, a parameter whose value is a list of values separated by commas, or even a custom command whose output lines are taken as items. Each item is available to the task code as an environment variable.

When all spawned tasks have finished, the foreach task ends as well and subsequent dependent tasks can be run. Nevertheless, there are several scenarios in which a pipeline developer can define multiple consecutive foreach tasks intended to iterate over the same collection of independent items. In these scenarios, when one iteration of a loop has finished, the corresponding iteration of the next loop could start without waiting for the entire previous loop to finish. Compi supports this type of interaction between foreach tasks by simply adding the ‘*’ prefix to the task name when declaring the dependency on the “after” attribute. For example, in a scenario where a pipeline must process a set of samples (e.g: “case-1”, “case-2”, “control-1”, “control-2”) by performing two consecutive operations: preprocess and analyze, the dependency of the “preprocess” task on the “analyze” task can be prefixed with ‘*’ (i.e., after =”*preprocess”) to tell Compi that iterations over the “analyze” loop could start when the corresponding iterations over the “preprocess” loop have finished.

Figure 3 shows an example of this, where two foreach tasks “preprocess” and “analyze” iterate over the same set of items (“samples”). The “analyze” task depends on the “preprocess” task, but at an iteration-level (after =”*preprocess”). Without the ‘*’ prefix, the “analyze” task will only start when the whole “preprocess” task has finished.

Pipeline execution and dependency management

One of the main features of Compi is the creation of a classic CLI for the entire pipeline based on its parameter specifications to aid users run the pipeline. This CLI is displayed when a user executes “compi run -p pipeline.xml –help”, which describes the Compi execution parameters and specification of each pipeline task. File S1 shows this CLI for the RNA-Seq Compi pipeline (https://www.sing-group.org/compihub/explore/5d09fb2a1713f3002fde86e2).

As Fig. 4 illustrates, the execution of Compi pipelines can be controlled using multiple parameters that fall in three main categories: pipeline inputs (i.e., the pipeline definition and its input parameters), logging, and execution control. In Fig. 4A, the “compi run” command receives the pipeline definition file explicitly, while the “–params” option indicates that the input parameters must be read from the “compi.params” file. On the other hand, in Fig. 4B the pipeline definition file is omitted and compi assumes that the pipeline definition must be read from a file named “pipeline.xml“ located in the current working directory. Also, pipeline parameters are passed on the command line after all the “compi run” parameters and separated by the ‘–’ delimiter.

Figure 3 Example of Compi pipeline using iterative foreach tasks.

Note the “*” character when the dependency of the “analyze” task on the “preprocess” task is defined. This way, the second foreach is binded to the first one, meaning that the two foreach tasks iterate over the same collection of elements and allowing Compi to start the execution of each iteration of the second foreach right after the corresponding iterations of the first foreach had finished.

Figure 4 Examples of Compi parameters to control how pipelines must be executed.

Compi parameters belong to three main categories: pipeline inputs (i.e., the pipeline definition and its input parameters), logging, and execution control (i.e., specify whith tasks must be executed).

Regarding logging options, both include the “–logs” option to specify a directory to save the standard (stdout) and error (stderr) outputs of each task along with the specific parameter values used in each execution. They are saved in three different files named with the name of the corresponding task as prefix (e.g., “task-name.out.log“,“task-name.err.log“, and “task-name.params“). To avoid unnecessary file creation, Compi does not save task outputs unless the “–logs” option is used. However, it is possible to specify which tasks should be logged using the “–log-only-task” or “–no-log-task” parameters. In addition to the task specific logs, Compi displays its own log messages during pipeline execution. These messages can be disabled by including “–quiet“ as shown in Fig. 4B. In contrast, in the example given in Fig. 4A, the “–show-std-outs” option forces Compi to forward each task log to the corresponding Compi output, which is very useful for debugging purposes during pipeline development.

The third group of options allows to control the execution of the pipeline. For instance, the “–num-tasks” parameter used in Fig. 4A sets the maximum number of tasks that can be run in parallel. It is important to note that this is not necessarily equivalent to the number of threads the pipeline will use, as some tasks may use parallel processes themselves. The “–abort-if-warnings” option, also used in the example in Fig. 4A, tells Compi to abort the pipeline execution if there are warnings in the pipeline validation. This is a useful and recommended option for pipeline testing during development, to avoid undesired effects that may arise from ignoring such warnings. A typical scenario that causes a warning is when the name of a pipeline parameter is found inside the task code, but the task does not have access to it, because it is not a global parameter nor defined in the set of task parameters.

One of the most notable features of the Compi workflow execution engine is that it allows a fine-grained control over the execution of the pipeline tasks. While many workflow engines only allow launching the entire pipeline or partially relaunching a pipeline from a point of failure, which can also be done in Compi using the “resume” command, Compi also allows launching sub-pipelines using modifiers such as “–from”, “–after“, “–until“ or “–before“. In the example shown in Fig. 4B, a combination of the “–from“ and “–until“ modifiers is used, resulting in the execution of all tasks in the path between “task-1” and “task-10”, including “task-1” and “task-10”. If the “–after“ and “–before“ are used instead, then “task-1” and “task-10” are not executed. A fifth modifier is “–single-task“, which allows only the specified task to execute and is not compatible with the other four modifiers. The Compi documentation (https://www.sing-group.org/compi/docs/running_pipelines.html#examples) includes several examples to illustrate how each of these options work.

Compi runs each task as a local command by default (Fig. 5A). This means that if a task invokes a certain tool (e.g., a ClustalOmega alignment running “clustalo -i /path/to/input.fasta -o /path/to/output.fasta”), this tool must be available either from the path environment variable or by including the absolute path to the binary executable. Since dependency management is always cumbersome, a special effort has been made to offer developers and users various alternatives to deal with this problem, which are explained below.

Figure 5 Different ways of executing tasks involving external dependencies in Compi.

The upper left quadrant shows the simplest case (scenario A), where the pipeline is run on a single machine and external dependencies must be available on the system. One way to manage external dependencies is by using containers like the one shown in the bottom left quadrant. While scenario C shows the case of a Compi application which runs in a container with all dependencies, scenario D shows a Compi application that uses a Docker image for running the external dependencies. Custom Compi runners can also be used to submit tasks into an HPC cluster, as scenario B shows. Finally, it is also possible to combine containerized execution of Compi tasks with Docker in a clustered environment by using a container orchestration system such as Kubernetes.

One way to address this issue is by means of a file with a custom XML runner definition, as done in the example shown in Fig. 4A with “–runners-config pipeline-runners.xml”. Individual runners are defined through a “runner“ element within a runners file, where the “task“ attribute is used to specify the list of tasks that the runner must execute. In this way, when a task identifier is assigned to a runner, Compi will ask the runner to run the corresponding task code (instead of running it as a local command). The usage of pipeline runners to handle dependencies allows Docker images to take responsibility for them (Fig. 5D). For instance, Fig. 6A shows a pipeline task named “align” that uses a tool (defined by the pipeline parameter “clustalomega”) that receives a file as input and produces an output. The runner defined in Fig. 6B for the same task runs the specified task code (available in the environment variable “task_code”) using a Docker image. The runner here is almost a generic Docker runner, and the key points are:

Figure 6 Use of custom Compi runners.

(A) Task of a Compi pipeline that invokes a command defined in the “clustalomega” pipeline parameter. This command must be available in the system. (B) Example of Compi runner for the “align” task that runs the task source code (available in the “task_code” environment variable) using a Docker image that has the command installed.

• First, the creation of a variable (“$envs”) with the list of parameters that must be passed as environment variables to the Docker container.

• Second, run the Docker image with the list of environment variables and mount the directory where the command has the input and output files (“workingDir” in this example).

Such a Docker runner would allow to follow an image-per-task execution pattern, where each task is executed using a different container image (Spjuth et al., 2018). An example of this execution pattern can be found in the GenomeFastScreen pipeline (https://sing-group.org/compihub/explore/5e2eaacce1138700316488c1), although in this case “docker run” commands are included in each task rather than provided in a runners file for the sake of simplicity. Following this image-per-task execution pattern, it is possible for a pipeline to use different versions of the same software or two tools that require different versions of some dependencies.

In addition, custom runners can also be used to submit pipeline tasks to a job scheduler such as SGE, Torque or SLURM in supercomputers or computer clusters (Fig. 5B). For instance, Fig. 7 shows a generic Slurm runner. Some srun parameters may need to be adjusted for each specific cluster and the “–export” parameter must be used to export all environment variables to the process to be executed, as the task parameters are declared as environment variables.

In the same way, when using Compi runners, it is also possible to combine containerized execution of Compi tasks with Docker in a clustered environment, using a container orchestration system such as Kubernetes (Fig. 5E).

Another way to achieve dependency management is by building a monolithic Docker image with Compi, the pipeline itself and all its dependencies (Fig. 5C). This topic is explained in “Reproducible Application Packaging with Docker”.

Finally, dependency management can be also delegated to external systems. For example, Compi allows the use of Conda/Bioconda packages seamlessly. Each task can use them simply by activating and deactivating the corresponding Conda environments before executing the specific commands. The Metatax pipeline (https://www.sing-group.org/compihub/explore/5d807e5590f1ec002fc6dd83) illustrates this. For instance, Fig. 8 shows the execution of a script (whose name is defined by the “validate_mapping_file” parameter, highlighted in bold) from the Qiime Bioconda package.

Figure 7 Generic Compi runner for submitting tasks into a Slurm system.

Figure 8 Dependency management in Compi pipelines using Conda/Bioconda packages.

In this example from the Metatax pipeline, the “validate_mapping” task activates the qiime1 environment to load the Qiime Bioconda package, executes a script from this environment, and finally deactivates the environment. Each task can use them by simply activating and deactivating the corresponding environments before executing the specific commands.

Reproducible application packaging with docker

Compi enables the creation of portable end-user CLI applications for pipelines that can be distributed as Docker images. As noted in the previous section, this is another way to deal with dependency management, as such Docker images contain all the dependencies required by the pipeline. Pipelines distributed in this way follow an image-per-pipeline execution pattern in which all tasks are executed using the same image container (Spjuth et al., 2018) (Fig. 5C), and can even be run using Docker-compatible container technologies such as Singularity.

The compi-dk command-line tool is provided to assist in the development and packaging of Compi-based applications into Docker images. Pipeline development starts with the creation of a new compi-dk project with the “compi-dk new-project” command, which creates a project directory and inizialites two template files: pipeline.xml and Dockerfile. After this, the definition of the pipeline can start by modifying the pipeline.xml template and the subsequent local testing (using the compi command). Also, it can be tested by building a Docker image (using compi-dk) and running the containerized pipeline. As for the latter case, when the compi-dk build command is executed on the project directory, a Docker image for the pipeline is created. This image contains the compi executable file and a specific pipeline.xml file, along with the pipeline dependencies as defined in the Dockerfile. Figure 9 shows the Dockerfile of the MINC Computer Vision pipeline for image classification based on Deep Learning (https://www.sing-group.org/compihub/explore/5d08a9e41713f3002fde86d5). The Dockerfile skeleton was automatically generated by compi-dk, being only necessary to add the “RUN” commands for the installation of “gluoncv” and “gnuplot” dependencies. In this regard, it is important to note that special attention was placed on the ability to derive a pipeline application image from any preexisting Docker image of interest (e.g., images with bioinformatics packages). For instance, the RNA-Seq Compi pipeline discussed above was created using the DEWE (López-Fernández et al., 2019) Docker image as a base image and the MINC Computer Vision was created using one of the Apache MXNet Docker images as a base image.

Figure 9 Docker image for the MINC computer vision pipeline (https://sing-group.org/compihub/explore/5d08a9e41713f3002fde86d5).

When working with a compi-dk project, it is also possible to create a Docker image for a pipeline that follows the image-per-task execution pattern. This is the case of the GenomeFastScreen pipeline (https://sing-group.org/compihub/explore/5e2eaacce1138700316488c1), in which, as explained above, a “docker run” command is included within each task instead of managing the execution of Docker using an external runner file. Regardless, most of the tasks within this pipeline run under external Docker images, thus following the image-per-task execution pattern. As the GenomeFastScreen pipeline itself is also distributed as a Docker image, it must be able to run Docker images as well (please refer to the pipeline documentation for details on how to do this).

Pipeline distribution via Compi Hub

Compi enables the creation of portable end-user CLI applications for pipelines that can be distributed as Docker images. As noted in the previous section, this is another way to deal with dependency management, as such Docker images contain all the dependencies required by the pipeline. Pipelines distributed this way follow an image-per-pipeline execution pattern in which all tasks are executed using the same image container (Spjuth et al., 2018) (Fig. 5C), and can even be run using Docker-compatible container technologies such as Singularity.

Once the pipeline development is completed, it can be released through Compi Hub to increase the visibility and benefit from the Compi Hub features. Pipelines can be registered using the Compi Hub web interface (https://www.sing-group.org/compihub) or using the “compi-dk hub-push” command. Compi Hub can store several versions of a pipeline, each of them associated to a pipeline.xml file where the specific pipeline version is defined. In addition, since Compi Hub does not store either the full source code (e.g., scripts included in the pipeline.xml as source files) or the Docker images themselves, pipeline publishers are encouraged to: (i) publish the source code (i.e., the compi-dk project) in public repositories such as GitHub or GitLab to allow users to re-build the project locally at any time; and (ii) push the corresponding Docker image to the Docker Hub registry so that users can pull the image and follow the instructions to run the pipeline application.

The Compi Hub website lists publicly available pipelines and gives access to all pipelines. When a pipeline is selected, the main pipeline information is displayed, including title and description, creation date, as well as links to external repositories in GitHub or Docker Hub. In addition, for each pipeline version, Compi Hub displays the following information:

• Overview: this section is headed by the pipeline DAG, generated in the backend using the “compi export-graph” command. Since it is an interactive graph, visitors can use it to navigate to each task description. Figure 10 shows the Metatax pipeline DAG (https://www.sing-group.org/compihub/explore/5d807e5590f1ec002fc6dd83). The DAG is followed by two tables, one containing the pipeline tasks and their associated descriptions, and a second one containing global parameters of the pipeline. Finally, this section encloses one table for each task with descriptions and specific parameters. It is important to note that all this information is automatically generated from the pipeline XML.

• Readme: this section shows the content of the README.md file when it is present in the compi-dk project. This file should be used to provide a comprehensive description of the pipeline, as well as instructions on how to use it.

• Dependencies: this section shows the content of the DEPENDENCIES.md file when it is present in the compi-dk project. We recommend that pipeline developers include this file with a human-readable description of the pipeline dependencies and the specific versions used to develop the pipeline.

• License: this section shows the content of the LICENSE file when it is present in the compi-dk project. We encourage pipeline developers to include this file in their compi-dk projects so that the terms of use of the pipeline are clear.

• Dataset: this section contains a list of datasets that can be used to test the different versions of the pipeline. It is shown when a pipeline publisher associates a test dataset with the displayed pipeline. In addition to the instructions given in the Readme section, we also recommend that pipeline publishers provide test datasets to help users test the pipelines themselves.

• Runners: this section displays a list of example runner configurations when they are present in the compi-dk project. Runner configurations must be stored as XML files within the “runners-example” directory of the project.

• Params: this section shows a list of example parameter configurations when they are present in the compi-dk project. Parameter configurations must be stored as plain-text key-value files in the “params-example” directory of the project.

As can be seen, Compi Hub was not designed to be a merely pipeline repository. We seek that developers accompany each pipeline with all the necessary information to ensure its portability and reproducibility by other researchers.

Figure 10 DAG of Metatax as shown in Compi Hub (https://www.sing-group.org/compihub/explore/5d807e5590f1ec002fc6dd83).

Compi Hub automatically generates it by running the “compi export-graph” command in the backend. For each, tasks are drawn with a dashed border to differentiate them from regular tasks.

Discussion

Workflow management systems play a key role in the development of data science processing pipelines in multiple fields, such as bioinformatics or machine learning, among others. There are multiple solutions and approaches to develop flexible, portable, usable, maintainable and reproducible analysis pipelines in an easy way. With Compi, we progressed from a state-of-the-art capable workflow management system to an entire application framework, focusing on the transition from pipeline development to end users and the community. In this sense, we have put special emphasis on providing pipelines with an advanced and automatically generated CLI, which is aware of the pipeline structure, parameters and tasks descriptions, to facilitate its adoption by final users. Moreover, Compi aids pipeline developers in creating all-in-one distributable Docker images, as well as share the pipeline in an online automatically documented hub where it will be available to the community. This combination with Docker has the added benefit that Compi projects can be built into runnable CLI applications using only text files (i.e., a “pipeline.xml” file, a Dockerfile, and other project files), allowing pipeline developers to use control version systems to keep track of their development. Compi pipelines can be executed in multiple computing layouts without any modification, from running natively on a single machine to a high-performance, fully containerized cluster environment, tailored to the needs of the end user. At design level, we have prioritized the use of well-known standards, such as XML, low intrusiveness and language agnosticism, targeting a broader user community. In this sense, we avoid defining pipelines via DSLs based on a specific programming language, forcing a scripting language to define each task code, or be coupled to specific dependency management systems, such as Python (Conda), R (Cran) or Java (Maven).

Unlike Snakemake, the Nextflow and SciPipe tools allow dynamic scheduling, that is, the ability to change the pipeline structure dynamically to schedule a different number of tasks based on the results of a previous step or any other parameter. Compi allows dynamic scheduling via (i) the “if “ attribute of tasks, which executes a command just before the task is about to run, allowing the task to be skipped dinamically, and (ii) foreach loops, which can take their iteration values from the output of a command, which is executed just before the foreach loop is about to run, allowing, for instance, to do more parallel iterations depending on the number of files generated by a previous task.

Compi, Snakemake, and Nextflow are language independent, allowing external scripts written in any programming language to be invoked.

Similar to Compi runners, Nextflow defines executors, which are the components that determine where a pipeline process runs and its execution is supervised. It provides multiple built-in executors to manage execution on SGE, SLURM, Kubernetes, and many others. In SciPipe, this can be achieved by using the Preprend field when defining processes, similarly to how Compi runners work.

Regarding containerization, as explained above, when pipeline tasks need to run in isolated containers, Compi users must include the corresponding command (e.g., “docker run”) in the task code or in a runner that tackles the execution of such tasks. Additionally, pipeline developers can create a Docker image for the entire pipeline so that all tasks are executed in the same container. In both cases, the developer must mount the paths to the input and output files of each task when using Docker. In this sense, Nextflow provides built-in support to run individual tasks or complete pipelines using Docker, Singularity or Podman images. In the case of Docker, Nextflow is able to mount the input and output paths automatically, since it is aware of the files needed by tasks. Similarly, Snakemake has built-in support to execute complete pipelines on Docker images and individual rules in isolated Conda environments. The SciPipe documentation does not provide information on how to containerize pipelines.

Logging is another important feature of workflow management systems, allowing pipeline users to see how execution went and determine causes of errors if necessary. SciPipe has been designed with special care on logging and collecting metadata about each executed task. Following a data-centric audit logging approach, SciPipe generates a JSON file for each output file that contains the full trace of the tasks that were executed to generate it. Nextflow provides a log command that returns useful information about a specific pipeline execution, and incorporates a “–with-report” option in the run command that instructs Nextflow to generate an HTML execution report that includes many useful metrics about a specific workflow execution. It also supports a “–with-trace” option in the run command that generates an execution trace file that contains useful information about each process executed as part of the pipeline (e.g., submission time, start time, completion time, CPU, and memory usage). As previously stated, the “compi run” command provides the “–logs” option to specify a directory to save the stdout and stderr outputs and the specific parameter values of each task execution in separated files prefixed with the corresponding task name.

Nextflow has nf-core, an environment with a dual purpose: to provide an online repository of Nextflow pipelines and to provide a command-line tool to interact with the repository and manage the execution of the hosted pipelines. Similarly, the Compi Hub repository allows users to discover and explore pipelines, and the compi-dk tool allows to push pipelines to the hub via command line. Snakemake does not provide a similar public repository, but a dedicated GitHub project exists (https://github.com/snakemake-workflows/docs).

After these considerations, and given the choices available, we believe that Compi may be a reasonable choice for researchers with CLI skills looking to create medium complexity pipelines without requiring them to learn new programming languages (e.g., to learn the Nextflow DSL or Go for SciPipe pipeline development). In this way, researchers would benefit from the common features of the workflow management engine that Compi offers without the need for much training. In addition, thanks to the auto-generated CLI, Compi would be the most suitable solution for those pipelines meant to be used by researchers as end-user applications. As the existing pipeline examples presented in the previous section demonstrate, pipeline-based applications developed in this way can easily be distributed as Docker images. Finally, to illustrate the way of creating Compi pipelines and the main differences with other workflow management systems, we have implemented a Nextflow example pipeline in Compi (File S2). This simple example, the description made in the previous section, and the public pipelines available on Compi Hub, will allow potential users to determine when Compi may be the most suitable option.

Conclusions

Compi is an application framework for developing pipeline-based end-user applications in bioinformatics and data science. Two Compi design principles are low intrusiveness and language agnosis, with the goal of covering a wide variety of scenarios and providing the most flexibility to pipeline developers. Noteworthy, Compi pipelines can be executed in multiple computing layouts without the need to modify the pipeline definition, from running natively on a single machine to a fully-containerized, high-performance cluster environment, tuned for the needs of the end user.

To complement the Compi workflow execution engine, we have also created Compi Development Kit (compi-dk) and Compi Hub. Thanks to the Compi Development Kit, pipelines can be packaged as self-contained Docker images that also include pipeline dependencies, and can be shared publicly with minimal effort using Compi Hub.

Future work include, among other tasks, the following issues: (i) improving the metadata section of the pipelines to allow the inclusion of more information (e.g., licensing, attributions, or custom information); (ii) including the possibility of creating stackable runners that gives more flexibility and power to customize task execution; and (iii) enhancing the logging reports generated by Compi.

Supplemental Information

Supplemental Information 1 Command-line interface for the RNA-Seq Compi pipeline

Click here for additional data file.

Supplemental Information 2 Implementation of a Nextflow example pipeline in Compi

Click here for additional data file.

SING group thanks the CITI (Centro de Investigación, Transferencia e Innovación) from the University of Vigo for hosting its IT infrastructure.

Additional Information and Declarations

Competing Interests

Author Contributions

Data Availability

The authors declare there are no competing interests.

Hugo López-Fernández and Daniel Glez-Peña performed the computation work, prepared figures and/or tables, authored or reviewed drafts of the paper, and approved the final draft.

Osvaldo Graña-Castro, Alba Nogueira-Rodríguez and Miguel Reboiro-Jato performed the computation work, authored or reviewed drafts of the paper, and approved the final draft.

The following information was supplied regarding data availability:

The Compi code is available at GitHub (https://github.com/sing-group/compi). The code of referenced Compi pipelines is available in CompiHub (http://sing-group.org/compihub).

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
