# Peer review of "Compi: a framework for portable and reproducible pipelines"

_PeerJ Computer Science, doi:10.7717/peerj-cs.593_

## Round 0.1 · original submission · Major Revisions

Both reviewers made a number of comments about the integration of Compi into the ecosystem of workflow management systems. Could you please compare an example workflow?

Also, could you make reference to the CWL - Common Workflow Language?

Best regards,
Robert

·

Basic reporting

Just a personal comment related to sentence beginning at line 216. There are many life science researchers who are circumstantial developers out there who are shy sharing their code, as they feel embarrassed about it. Therefore, most of those researchers are not used to version control systems yet.

First sentence in line 477 should be rewritten, so it is not misunderstood. It took me re-reading it at least twice, carefully, in order to get its meaning.

Experimental design

Line 485. I cannot agree with sentence "According to its documentation, Snakemake only allows the execution of external scripts written in Python, R, R Markdown, and Julia". If you have a look at https://snakemake.readthedocs.io/en/stable/snakefiles/rules.html and at example at https://snakemake.readthedocs.io/en/stable/tutorial/basics.html#step-1-mapping-reads , you can see there that it supports "shell" command line run. Snakemake supports rules where some python code can be embedded through "run" keyword, similar to Nextflow functionality, where groovy code can be run using "exec" keyword https://www.nextflow.io/docs/latest/process.html#native-execution .

Validity of the findings

No comment

Additional comments

I have read the manuscript, and also the related documentation, scattered among the repository and the different web pages. I have both comments, issues, questions and suggestions:

Line 209 of the manuscript: I guess there is a typo, as "Compo Hub" should be "Compi Hub".

Related to Compi Hub and its links in the manuscript, I guess I have found a bug in Compi Hub behaviour (2021-02-22, Firefox browser). I tried both links at line 269 ( https://www.sing-group.org/compihub/explore/5d09fb2a1713f3002fde86e2 ), at line 357 ( https://www.sing-group.org/compihub/explore/5d807e5590f1ec002fc6dd83 ) and at line 390 ( https://www.sing-group.org/compihub/explore/5e2eaacce1138700316488c1 ), but they did not work. I went to Compi Hub in order to find the referred RNA-Seq, Metatax and GenomeFastScreen pipelines, and I found then at "compi-rnaseq-pipeline", "metatax" and "pss-genome-fs" with the very same links. Clicking there it worked, but copying and pasting the links in a sibling tab or window did not work. The same happens to https://www.sing-group.org/compihub/explore , which is the URL https://www.sing-group.org/compihub/ redirects to.

I have perceived that the different sites and documentation are a bit isolated among themselves.

* I have missed both in http://sing-group.org/compi/docs/ and https://github.com/sing-group/compi/blob/master/README.md a reference to the download area at http://sing-group.org/compi/#downloads , as well as installation instructions (at least, the easy ones).

* I have also missed in http://sing-group.org/compi/docs/ a reference to https://github.com/sing-group/compi/ .

* Also, I have missed in https://github.com/sing-group/compi/ an INSTALL.md or similar explaining the steps to build compi on different platforms or architectures (which could be needed by macOS users, for instance). This comment is related to sustainability of a workflow written in Compi .

* Releases at https://github.com/sing-group/compi/releases could be enriched with the installers you are already providing at http://sing-group.org/compi/#downloads .

I have been studying the custom runners behaviour, and they are a very interesting feature, albeit quite synchronous. For instance, Slurm runner example https://www.sing-group.org/compi/docs/custom_runners.html#generic-slurm-runner is using the synchronous executor srun. Scenarios which require implementing custom runners which involve asynchronous checks to some resource are harder to implement. Could you include in the online documentation an advanced example of "polling" in a custom runner?

As it is distilled from line 429 to line 438, and from line 444 to line 446 good (scientific) software practices should be encouraged. So, I propose you that "compi-dk new-project" command should create initial LICENSE.md, README.md and DEPENDENCIES.md files, as well as a "runners-example" directory with a couple of the more common custom runners, and "compi-dk build" should check their existence (and give a warning when they are not found).

I have also several questions and suggestions to the authors, related to future developments of compi and Compi Hub:

Have you thought on an expandable workflow metadata system for Compi workflows? Currently it is a key / string value system bound to tasks, so it is not possible to have metadata attached to the whole workflow, its inputs and outputs.

I have a question which might be related to previous suggestion. How the authors of a workflow can embed the details of its licence and attributions? Currently that information must be either it is derived from the git repository of the workflow or it should be embedded as a comment in the workflow, as metadata system does not support it.

Do you have sub-workflow support in Compi's roadmap? My question is more related to common sub-workflow patterns reuse than composing a meta-workflow.

Execution provenance. A way to report what happened in an execution would be generating a parse-able report with all the parameters, both explicit and implicitly set, along with additional information and metrics of the workflow execution and the custom runners.

Should workflow semantic versioning be only recommended or enforced? My question is two-fold, both philosophical and related to that the there is no restriction on the version string at XML Schema definition.

Although custom runners are a very powerful feature, they have greater potential. So, I recommend having in the roadmap stackable custom runners, so stacks of custom runners would allow easy combinations of, for instance, Slurm+singularity or SGE+conda environment activation.

An additional suggestion about custom runners is allowing them to use task context annotations or variables. In that way, custom runners would be more reusable. For instance, a runner needing a conda environment precondition with an specific package installed could learn which conda package should be installed from that received metadata. Or using different containers for different tasks, based on an annotation provided by the task.

Last suggestion involving custom runners is having in Compi Hub a way to upload custom runner templates, so researchers can reuse successful patterns.

Reviewer 2 ·

Basic reporting

This manuscript describes the features of Compi- an application framework to develop end-user, pipeline-based applications with emphasis on user interface generation, application packaging, with compi-dk, and application distribution provided through a public repository of Compi pipelines, named Compi Hub.
I find it quite interesting that the authors have developed an XML-based platform for creating workflows and I also had a look at the resources for Compi at (i) http://www.sing-group.org/compi/ and (ii) https://sing-group.org/compihub/explore and they seem to be pretty well-documented.
However I have some reservations, namely:
(i) The paper is written more like a user manual instead of an application paper - there are also some grammatical mistakes. e.g. line 117 has an incomplete sentence "Fourth, for security reasons, since Compi is intended to run pipelines defined by third party programmers." Similarly there are more.
(ii) The authors have not mentioned CWL - Common Workflow Language, at all. I would like to draw their attention to the fact that CWL (https://github.com/common-workflow-language) is a very mature workflow language, being used by many bioinformatics communities. They can check https://www.sevenbridges.com/cwl-seven-bridges-platforms/
(iii) Given that they have compared Compi with Nextflow, I would expect that they take a simple workflow as an example and demonstrate how it is defined in both in NextFlow and Compi so that readers can see the advantages of Compi compared to NextFlow.While there already exist NextFlow and CWL as two major workflow languages, readers must be convinced why they should use Compi.

Experimental design

No comment

Validity of the findings

No comment

---

## Round 0.2 · accepted · Accept

Both reviewers agree that you have carefully addressed all their comments. If you wish, you can consider the additional notes of reviewer 1 in the proof stage. Congratulations and good luck with the promotion of Compi.

·

Basic reporting

I have checked the bibliographic reference you have provided to CWL, it is a nice one (and one of the first citing CWL in some way, as I have learnt from https://www.zotero.org/groups/2294829/cwl/items/FZ7GRJWP/library). I recommend to keep this one, and also adding as reference what it is requested at https://www.commonwl.org/#Citation

Peter Amstutz, Michael R. Crusoe, Nebojša Tijanić (editors), Brad Chapman, John Chilton, Michael Heuer, Andrey Kartashov, Dan Leehr, Hervé Ménager, Maya Nedeljkovich, Matt Scales, Stian Soiland-Reyes, Luka Stojanovic (2016): Common Workflow Language, v1.0. Specification, Common Workflow Language working group. https://w3id.org/cwl/v1.0/ doi:10.6084/m9.figshare.3115156.v2

which is eligible as "Gray Literature" (https://peerj.com/about/author-instructions/#reference-section)

Experimental design

No comment

Validity of the findings

No comment

Additional comments

Meanwhile I was reading the detailed description of Compi Hub (around line 421) and checking the claims in the browser, I have missed in the overview tab of Compi Hub a legend explaining the meaning of each one of the possible symbols in the graph. For instance, I had to have a look at the XML of the workflow https://www.sing-group.org/compihub/explore/5d807e5590f1ec002fc6dd83#overview to realize that "join_pe" was a foreach, not an optional step.

Also, when I tried finding some keywords related to the command-line in the documentation at http://www.sing-group.org/compi/docs/ I have realized that there is some issue, as it did not work. I have tried both on Firefox and Chrome

Reviewer 2 ·

Basic reporting

No comment

Experimental design

No Comment

Validity of the findings

No Comment

Additional comments

Based on the revised manuscript (in track-changes version) and the rebuttal letter, I am happy to note that the authors have really improved the manuscript and addressed my earlier comments.
They have discussed CWL as another well-known workflow framework, as well as provided an equivalent workflow in Compi for one in Nextlfow (in Supplementary 2) and corrected the English grammatical mistakes.
Hence I now find the manuscript acceptable for publication.